# Long-Term Multi-Sensory Gamma Stimulation of Dementia Patients: A Case Series Report

**DOI:** 10.3390/ijerph192315553

**Published:** 2022-11-23

**Authors:** Amy Clements-Cortes, Lee Bartel

**Affiliations:** Faculty of Music, St. George Campus, University of Toronto, 80 Queens Park, Toronto, ON M5S 2C5, Canada

**Keywords:** dementia, gamma stimulation, case series, cognition, mood

## Abstract

Dementia prevalence is increasing globally, and symptom management and treatment strategies require further investigation. Music-based interventions have demonstrated some efficacy with respect to quality of life and symptom reduction, though limited with respect to cognition. This study reports on three case studies where the use of gamma stimulation over one year contributed to maintenance of cognition and increases in mood for participants with Alzheimer’s disease or mild cognitive impairment. Auditory stimulation with isochronous sound at 40 Hz was delivered to participants via a commercially available vibroacoustic chair device five times per week for 30 min with assistance from caregivers. Further research is needed to assess the integration of this therapy in the overall care for persons with dementia.

## 1. Introduction

On September 2021, the World Health Organization (WHO) reported that around 55 million people worldwide are living with some form of dementia and that this number is expected to grow to 78 million by 2030 [1]. The WHO also observed that there is currently no cure for dementia and that medications and disease-modifying therapies that have been developed to date have limited efficacy. One therapy that is thought to have potential application in dementia-related care, especially for Alzheimer’s disease (AD), is music intervention.

An examination of the recent systematic reviews of the effects of music interventions on AD and dementia reveals that there are some arousal effects most clearly seen in verbal fluency and some calming effects seen as a reduction in anxiety, depression, and agitation, with these symptom reductions having a positive effect on quality of life and social interactions [2,3,4]. However, evidence that music interventions including music therapy have a positive effect on cognitive functions is scarce, with little evidence that there is a restorative or lasting effect. A recent fine-grained meta-analysis of music-based interventions with people with mild cognitive impairment and dementia parsed specific cognitive effects and found a significant improvement in general cognitive function, in episodic memory, and executive function but no significant improvement in processing speed. The meta-analysis found no studies that assessed working memory, short-term memory, or attention [5].

What gives many people hope regarding music interventions is that individuals with dementia often respond very favourably to music, particularly those with old music memories seemingly intact and whose music-making abilities are still functional [6]. Since music involvement at least at the listener level is societally pervasive, the cognitive functions related to music may result from a domain-specific cognitive reserve, as occurs with other artistic abilities [7,8,9]. Although the use of these preserved abilities may have a quality of life effect, it does not necessarily have a therapeutic effect on cognitive function, particularly on working and short-term memory.

The recent meta-analysis of music interventions for people with mild cognitive impairment and dementia specifically excluded studies that used interventions “focusing on music or sound as vibrotactile stimulation, single frequency sound, or sound for its vibratory effect” [5]. When music is not studied just as a cognitive phenomenon but rather as a neuro-modulatory rhythmic sensory stimulant, new possibilities emerge, and new outcomes can be investigated.

The foundational mechanisms for a possible mitigating or restorative effect on AD and dementia have been explored in conjunction with neuro-modulatory sensory stimulation with animals [10]. The research by Iacarrino et al. found that visual stimulation in the gamma zone at 40 Hz reduced inflammation, increased microglia activity, reduced amyloid beta and tau levels, and increased blood flow to the brain—all correlates of AD. Results also showed improved cognitive performance. A human clinical study and a case report at the same time found significant improvement in cognitive function using vibrotactile and auditory stimulation at 40 Hz [11,12,13]. A study in Italy produced similar positive results with a 40 Hz vibroacoustic stimulation [14]. Most recently, Martorell et al. [15] found further animal evidence for the effectiveness of 40 Hz gamma auditory and visual stimulation. Campbell et al. [16] recently pointed to the positive role of low frequency vibro-stimulation in dementia care. However, none of these studies have looked at the long-term effects as this report of cases does.

The selection of 40 Hz as the stimulus frequency is based on several factors. Studies have shown that, with the onset of AD, patients have reduced gamma activity, particularly at 40 Hz [17,18,19]. Gamma activity is important for cognition and memory and is elucidated in Clements-Cortés [11]. Efforts to drive a gamma response with sensory stimulation have shown greater efficacy when used alongside several sensory stimuli [11,15]. Studies have not yet explored whether the synchronization of afferent signals from multisensory (e.g., eye, ear, skin) stimulation is a factor in effectiveness.

Comorbidities of AD and dementia frequently include: (1) mood disorders such as depression and anxiety, (2) pain, and (3) sleep disorders. Music and low frequency sound stimulation have shown positive effects with these conditions [20,21,22,23,24].

## 2. Study Design and Instrumentation

This paper provides an account of three participants with dementia and their caregivers’ experiences of the use of multi-sensory gamma stimulation. The case report (CARE) guidelines and checklist were followed in preparing these reports [25]. Primary participants included three females (two diagnosed with Alzheimer’s disease (AD) and one with mild cognitive impairment (MCI)) ranging in age from 71–82. Secondary participants included the spouses or partners of the primary participants. The purpose of this study was to evaluate the therapeutic application of sound delivered through specialized equipment on the arousal of activity in the brain in persons with dementia.

The primary participants received auditory stimulation with isochronous sound at 40 Hz (160 Hz sine wave carrier tone modulated with 100% squelch at 40 Hz) and tactile stimulation at 40 Hz (sine wave). The multi-sensory stimulation was delivered through the commercially available Sound Oasis VTS 1000 [26] with the auditory stimulation delivered through speakers on either side of the participant’s head and the vibrotactile stimulation created with a low frequency transducer in the middle of the participant’s back. The recommended volume level of the auditory stimulus and the intensity of the vibration were demonstrated by the co-investigator; however, participants were allowed to adjust this for their comfort. The 30 min audio/vibrotactile track created by the co-investigator for the treatment was supplied to the participants as an MP3 file that was played through the vibro-music device from a computer or MP3 player.

Primary participants used the vibrotactile device five times per week for 30 min per day for one year and adhered to the treatment schedule as reported by secondary participants. Secondary participants were trained to administer the multi-sensory gamma treatment with the vibro-music device in an online session over Zoom with one co-investigator.

Primary participants were assessed three times: (1) before the multisensory treatment began (pre), (2) after eight weeks of treatment, and (3) after one year (post). At each point, cognitive function was assessed with two instruments: (1) the standardized St. Louis University Mental Status Examination (SLUMS) [27] and (2) a set of research-created questions (see Appendix A). Moreover, at each point, additional outcome data were gathered on the primary participants’ mood, anxiety, energy, pain, cognition, clarity, alertness, and quality of life. These data were obtained by having the secondary participant (caregiver) rate their perception of the primary participant’s state on a 1–5 Likert scale where 1 is the low end and 5 is the high end (see Appendix B). A third category of qualitative data was obtained for each primary participant at each assessment point through interviews of the secondary participants (caregivers) (See Appendix C).

The SLUMS examination tool was selected because it is designed to be sensitive to those with a mild cognitive impairment as well as AD. The test consisted of 11 items (30 points), and the assessment included attention, immediate, intermediate, and delayed recall, orientation, numeric calculation, visual spatial, and executive functions. Although the Mini-Mental State Examination (MMSE) is used more commonly, research comparing the MMSE and the SLUMS has shown that the two tests have similar sensitivities, specificities, and area under the curve, but that SLUMS is more sensitive to mild neurocognitive disorder [28].

## 3. Case Narratives

### 3.1. Case 1

Case 1 (C1) was a 77-year-old female diagnosed with AD in 2017. She had no other major health conditions other than digestion and constipation. Medications included Galantamine, a drug designed to prevent the breakdown of acetylcholine, and thereby decreasing AD symptoms, and Venlafaxine, a drug used to treat depression and anxiety, a low-dose aspirin (80 mg), and natural supplements. At the onset of diagnoses, she displayed increasing anxiety, depression, and sleep issues. She and her husband walked daily, watched television, and she began to require assistance with completing steps to get dressed or engage in making meals. She started deferring to her husband and speaking less when engaging with others or at social gatherings. She began this study in June 2021 and completed it June 2022.

Pre-study C1 had a SLUMS score of 16/30 and scored 10/11 on the personalized questions. After 8 weeks, the SLUMS score increased to 17/30 and to 19/30 at the one year mark. The scores on the personalized questionnaire were 10/11 and 11/11 at the eight week and one year points, respectively. At the eight week and one year mark, C1 was taking the same medications; however, venlafaxine increased to 100 mg from 75 mg. She was experiencing further digestive issues but otherwise no changes in overall health. She used the vibro-music device each day in the afternoon around 4 pm. Her spouse reported that she was calmer when using the device and that her cognition for the most part has remained the same over the one-year period. She continued to walk and watch television as other ways of maintaining cognition. Anxiety increased at the 8 week mark from pre-study but then returned to a similar level post-study. Likert scale rankings by her spouse for mood, anxiety, energy, cognition, clarity, alertness, and quality of life remained fairly consistent, as demonstrated in Table 1 (Range is 0–5, with 1 for example being no anxiety to 5 being high anxiety.

### 3.2. Case 2

Case 2 (C2) was an 82-year-old female diagnosed with AD in 2019, osteoporosis in her 70s, and depression from early adulthood with no other comorbid health conditions. She was taking mirtazapine for mood and melatonin to help with sleep. These medications remained consistent over the course of one year and adendronic acid and fosamaz were added for osteoporosis after 6 months from study onset. Upon AD diagnosis, her spouse reported that she began having difficulty with navigating tasks and planning such as loading the dishwasher. Depressed mood ensued the formal diagnosis, and there was a phase wherein she did not recognize her husband. She had always been active and enjoyed reading; however, her activity level decreased, and she no longer read and was unable to initiate conversation with others; however, could engage in conversation when prompted. Her spouse attempted to engage her in walking and listening to music, but she was often unmotivated. She began the study in August 2020 and completed it in September 2021.

SLUMS scores for C2 were 13, 15, and 13, respectively, pre-, 8 weeks, and post-study. The scores on the personalized questionnaire were 8/11, 10/11, and 10/11 respectively. At the 8 week mark, C2 and her husband confirmed that she was adhering to the treatment 5 times a week, with no fixed time of day, but typically in the early evening. He also reported she was more alert and involved her in exercise in addition to encouraging daily walking and listening to music. At the one year mark, C2’s spouse reported that her health remained stable, with a slight decline in cognition and further progression of the osteoporosis. She was more social and was frequently interacting independently with others. There was one three-week gap in her adherence to the treatment in December when they traveled to visit family. Likert scale reports for the 7 outcome measures are noted in Table 2 and are reflective of increases in mood, cognition, quality of life, as well as decreases in anxiety and pain, with the other outcome measures remaining consistent.

### 3.3. Case 3

Case 3 (C3) was a 71-year-old female who was diagnosed with mild cognitive impairment (MCI) in Spring 2019. At onset, she was taking Synthroid for hyperthyroidism and cholesterol medication. Her medication remained consistent over the course of the study, and she had no other major health diagnoses or issues. At the onset of the MCI diagnoses, she could not retain new information, such as, for example, what she ate for lunch or the date. She also began to withdraw in social settings, anxiety increased, and she was less confident in performing the activities she typically did with her partner. Together, the couple would walk, watch TV, garden, and sometimes cook together. C3 mostly conversed with her immediate family, and their circle of friends was limited as a result of the COVID-19 pandemic restrictions. C3 began the study in November 2020 and completed it in December 2021.

SLUMS scores were 21, 23, and 21, respectively, pre-, 8 weeks, and post-study. She was also able to answer the personalized questions, consistently scoring 11/11 at each study interval. The couple reported adhering to the treatment and using the vibro-music device 5 times a week, typically after breakfast. The intensity of the vibration was reduced at the 8 week mark as C3 sometimes felt that the vibration was ‘strong’. This change was well received, and no further changes were made until the study’s completion. At 8 weeks, anxiety remained an area of concern with respect to worrying about memory decline. C3 described an increased feeling of hope and that she felt her overall health had improved. Energy was sometimes a concern and, during errands, C3 tended to become fatigued faster.

Short-term memory issues remained at the one year point; however, C3’s geriatric psychiatrist stated that she was doing well. Her spouse noted a “remarkable improvement in [cognition and overall health] since they started using the device”. Levels of brightness, mental health, and increased overall health were reported again as having improved. She was also motivated to participate in new experiences as she had before the diagnosis. Table 3 Likert scale ratings indicate increases in mood, energy, decreases in pain, and a slight decrease in cognition and a marked decrease in pain. Pain was not discussed as a significant issue and is likely an outlier here.

## 4. Discussion

Examination of the SLUMS scores (Figure 1) across the three time points shows that Case 1 improved in the first 8 weeks and continued to improve through the year. Cases 2 and 3 showed improvement after the first 8 weeks but then, by the end of the year, were at the same level as at the start of the year-long study. The gain in the short term from multi-sensory gamma stimulation is consistent with a previous study by the investigators [11,13], and the cessation of deterioration in all three cases is consistent with a previous case over three years [12]. With AD and dementia in general, it has been established that patients with these conditions continue to deteriorate on an annual basis [29,30].

An examination of the Likert Scale ratings of mental states by the caregivers shows that each of the three cases experienced an improvement in mood within the first 8 weeks and that they continued to improve for two of the cases through the year. See Figure 2 Likert Scale Scores for 3 Cases The effect of multi-sensory gamma stimulation on mood observed here is consistent with previous research [20,21]. The caregivers’ rating of cognitive function is somewhat inconsistent with the SLUMS scores obtained. In Cases 2 and 3, there is a noteworthy improvement reported in quality of life.

From the caregiver interviews, C1 and her spouse felt that the vibro-music device was beneficial in helping maintaining cognition and that she was “more perky”. They stated that there were no adverse effects from using the vibro-music device and that it became part of their routine. C1’s spouse said she was doing well overall. C2 and her spouse felt that the VTS device was beneficial in maintaining cognition and that, while they were unsure of the specific benefit, they were going to continue using the device as part of their daily routine. C3 and her partner were pleased with the outcomes from using the vibro-music device; they both described positive changes and that they would continue using this to support C3’s overall management of MCI. If we look across the three cases, multi-sensory gamma stimulation was well tolerated and, in fact, enjoyed.

There are other sources of stimulation for persons with AD including Snoezelen multi-sensory environments (MSE) which provide auditory stimulation and a range of sounds such as waterfalls and other sounds in nature [31]. Collier and colleagues (n.d.) found that exposure to MSE for persons with moderate to severe dementia resulted in significant motor performance improvements [32]. However, it must be noted that these other MSEs are not intended as specific neuromodulatory stimuli in the same way that this study demonstrates.

It is important to note that a small selection of illustrative and instructional cases, as in this report, has limitations, which include the fact that the participants did not present with complicated comorbidities which might impact their ability to engage in sound stimulation such as a hearing impairment. A strength of each case presented is the support of their spouse, who were instrumental in ensuring that participants adhered to the treatment schedule and likely also motivated or encouraged their participation. Since the cases in the series reported here have general common features including a diagnosis of dementia and received the same treatment over the same time period, conclusions can be drawn about the effect of the multi-sensory stimulation and cognitive function in these cases. However, more fine-grained variability analysis among the cases is not justified nor appropriate. Such a purpose would require a controlled study with a larger population. This needs to be performed in the future.

## 5. Conclusions

Dementia is prevalent and increasing globally. A holistic approach to care is required to ensure symptom management and the slowing of disease progression. Music-based interventions have demonstrated some improvement in symptoms and quality of life but limited impact on cognition. This study reported on the positive experiences of three primary participants with AD or MCI and their caregivers’ experiences of using a vibro-music device to deliver vibrotactile body stimulation from a low frequency transducer and a 40 Hz sine wave and simultaneously auditory stimulation with a 160 Hz carrier frequency isochronously modulated at 40 Hz, demonstrating the maintenance of cognition over a year-long period. The results are promising and point to a need for continued research.

## Figures and Tables

**Figure 1 ijerph-19-15553-f001:**
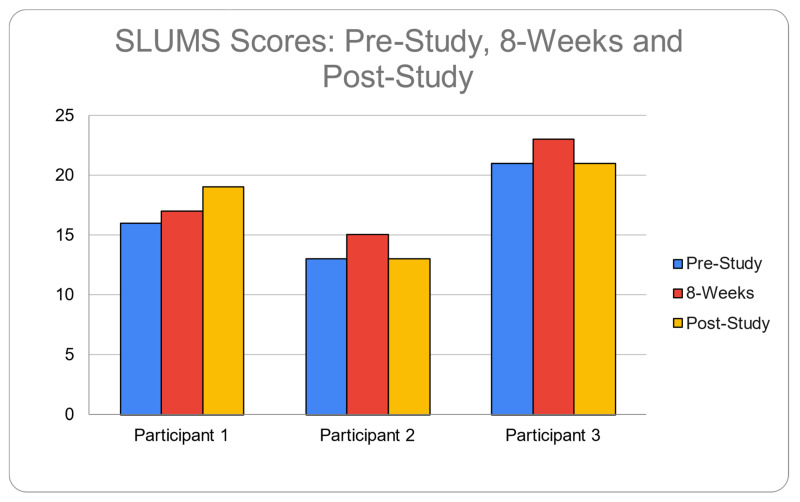
SLUMS Scores for 3 Cases.

**Figure 2 ijerph-19-15553-f002:**
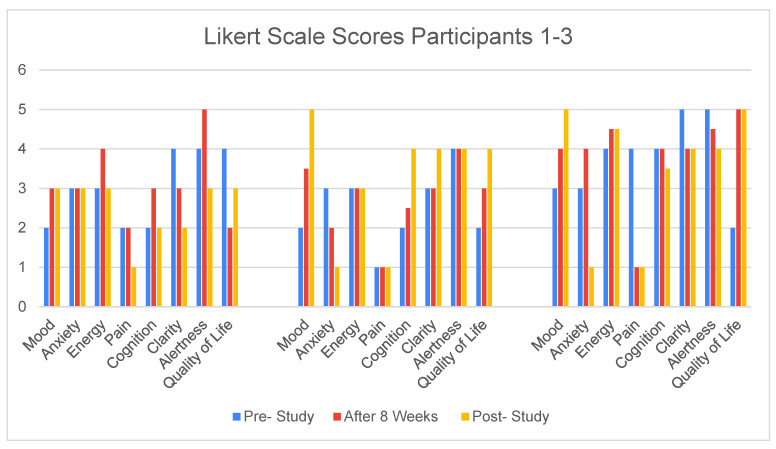
Participants’ Likert Scale Scores.

**Table 1 ijerph-19-15553-t001:** Case One: Likert Scale Ratings.

Measure	Pre-Study	After 8 Weeks	Post-Study
Mood	2	3	3
Anxiety	3	3	3
Energy	3	4	3
Pain	2	2	1
Cognition	2	3	2
Clarity	4	3	2
Alertness	4	5	3
Quality of Life	4	2	3

**Table 2 ijerph-19-15553-t002:** Case Two: Likert Scale Ratings.

Measure	Pre-Study	After 8 Weeks	Post-Study
Mood	2	3.5	5
Anxiety	3	2	1
Energy	3	3	3
Pain	1	1	1
Cognition	2	2.5	4
Clarity	3	3	4
Alertness	4	4	4
Quality of Life	2	3	4

**Table 3 ijerph-19-15553-t003:** Case Three: Likert Scale Ratings.

Measure	Pre-Study	After 8 Weeks	Post-Study
Mood	3	4	5
Anxiety	3	4	1
Energy	4	4.5	4.5
Pain	4	1	1
Cognition	4	4	3.5
Clarity	5	4	4
Alertness	5	4.5	4
Quality of Life	2	5	5

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
