# Peer review of "Long-Term Multi-Sensory Gamma Stimulation of Dementia Patients: A Case Series Report"

_ijerph, 2022, doi:10.3390/ijerph192315553_

Round 1

Reviewer 1 Report

As a qualitative study, it is very incomplete and poor quality.  Its not quality study by manuscript stucture and presented material. Baseline demographic characteristics, deeper interpretation and monitoring data, analysis not available. I can assume that in the first two clinical cases, the clinical response could have been increased by the psycho drugs used.

Author Response

We appreciate your review, but without some suggestions on how to improve the introduction or cited references we are unsure how you would like us to amend the manuscript. Medications for each of the cases presented were noted and we thank you for this comment, however these medications are also not known to have a wide or significant impact in decreasing cognitive decline if not paired with other stimulation.

Reviewer 2 Report

The paper is well written, with an adequate structure and good explanation of the whole process and methodology. The researchers describe clinical history, related to treatment where an original therapeutic approach is evidenced, and they have made a correct interpretation of the results obtained, with figures and tables supporting their explanations. Authors offer good comparisons with newer studies and contributing to empirical knowledge.

The topic addressed is of high social relevance and provide an interesting research question. Since its inception in the 1980s, vibroacoustic therapy help successful treatment approach for individuals with disabilities and special needs. Other researches have found it effective for reduction of stress, depression, mood disorders and anxiety, central nervous system stimulation or enhanced communication skills.  I really encourage them to continue in this line.

However, the author must specify the strengths and limitations in the approach to the cases and the importance of each limitation. Previous studies have already indicated that gamma brain waves thought to be involved in attention and memory because they are diminished in Alzheimer's disease. Moreover, authors did not present patients with unique characteristics or with unusual medical problems, such as people with hearing loss who differ perception of sounds.

 Some minor considerations can be made:

-         - I think that the authors should not have omitted their specific objectives.

-        -  Please state more clearly how was the informed consent obtained in legal incapacitated participants? This is a mandatory step in clinical research, so please provide more information about how you handled this issue.

-         - Please, add The CARE Guidelines: Consensus-based Clinical Case Reporting Guideline Development reference in the text.

-        - I recommend the authors to consider including Snoezelen® Multi-Sensory Environments references, in the discussion section, because this rooms give a complete and intricate experience and offer a great selection of sensory solutions for people with dementia.

Author Response

-         - I think that the authors should not have omitted their specific objectives.

Purpose of the study added. Limitation and strength added.

-        -  Please state more clearly how was the informed consent obtained in legal incapacitated participants? This is a mandatory step in clinical research, so please provide more information about how you handled this issue.

Information added on informed consent.

-         - Please, add The CARE Guidelines: Consensus-based Clinical Case Reporting Guideline Development reference in the text.

CARE guideline reference was added in text and in the reference list.

-        - I recommend the authors to consider including Snoezelen® Multi-Sensory Environments references, in the discussion section, because this rooms give a complete and intricate experience and offer a great selection of sensory solutions for people with dementia.

Two reference for the Snoezelen were added.

Reviewer 3 Report

In this study, the authors reported three case studies where the use of gamma stimulation over one year contributed to maintenance of cognition and increases in mood for participants with Alzheimer’s disease or Mild Cognitive Impairment. In this study, auditory stimulation with isochronous sound at 40 Hz was delivered to participants via a commercially available vibroacoustic chair device five times per week for 30 minutes with assistance from caregivers. These findings are interesting, but I have several concerns listed below.

1) Is multisensory stimulation superior to auditory or somatosensory stimulation alone? If yes, please explain why.

2) The authors should discuss whether the synchronization of auditory and somatosensory stimulation is necessary.

3) Why the stimulation was given at gamma frequency? How about lower or higher stimulation frequencies?

4) The authors should at least discuss some control conditions. For example, if continuous stimulation was given, instead gamma frequency stimulation, what would be the effects?

5) The authors should discuss the variability among the effects in the three patients.

Author Response

Reviewer 3

In this study, the authors reported three case studies where the use of gamma stimulation over one year contributed to maintenance of cognition and increases in mood for participants with Alzheimer’s disease or Mild Cognitive Impairment. In this study, auditory stimulation with isochronous sound at 40 Hz was delivered to participants via a commercially available vibroacoustic chair device five times per week for 30 minutes with assistance from caregivers. These findings are interesting, but I have several concerns listed below.

1) Is multisensory stimulation superior to auditory or somatosensory stimulation alone? If yes, please explain why.

2) The authors should discuss whether the synchronization of auditory and somatosensory stimulation is necessary.

3) Why the stimulation was given at gamma frequency? How about lower or higher stimulation frequencies?

These three points have been addressed in a new added paragraph as follows:

            The selection of 40Hz as the stimulus frequency is based on several factors. Studies have shown that with the onset of AD, patients have reduced gamma activity particularly at 40Hz [X1 X2, X3].  Gamma activity is important for cognition and memory and is elucidated in Clements-Cortés [11].  Efforts to drive a gamma response with sensory stimulation have show greater efficacy with the simultaneous use of several sensory stimuli [11,15]. Studies have not yet explored whether synchronization of afferent signals from multisensory (e.g., eye, ear, skin) stimulation is a factor in effectiveness.  

4) The authors should at least discuss some control conditions. For example, if continuous stimulation was given, instead gamma frequency stimulation, what would be the effects?

This question may be interesting but is not of direct relevance to the current study since the question is whether long term stimulation with 40Hz gamma is effective with AD.

Bernhard Ross, Simon Dobri, Shahab Jamali, Lee Bartel, Entrainment of somatosensory beta and gamma oscillations accompany improvement in tactile acuity after periodic and aperiodic repetitive sensory stimulation. International Journal of Psychophysiology,Volume 177, 2022, Pages 11-26.

https://doi.org/10.1016/j.ijpsycho.2022.04.007

See also the following for a stimulus that was specifically 40Hz and one that was more wide-band.

https://journals.plos.org/plosone/article?id=10.1371/journal.pone.0212021

5) The authors should discuss the variability among the effects in the three patients.

The following has been added in the discussion:  Since the cases in the series reported here have general common features including a diagnosis of  dementia and received the same treatment over the same time period, conclusions can be drawn about the effect of the multi-sensory stimulation and cognitive function in these cases. But, more fine grained variability analysis among the cases is not justified nor appropriate. Such a purpose would require a controlled study with a larger population. This needs to be done in the future.  

Round 2

Reviewer 1 Report

This research is relevant and may imply a wider use of non-medical measures for AD patienrs in the future?